# Hybrid Mamdani Fuzzy Rules and Convolutional Neural Networks for Analysis and Identification of Animal Images

**Hind R. Mohammed** [1,*] **and Zahir M. Hussain** [1,2]

1   Faculty of Computer Science and Mathematics, University of Kufa, P.O. Box 21, Najaf 540011, Iraq; z.hussain@ecu.edu.au
2   School of Engineering, Edith Cowan University, Joondalup, WA 6027, Australia
*   Correspondence: hindrustum.shaaban@uokufa.edu.iq; Tel.: +964-78-1098-6626

**Abstract:** Accurate, fast, and automatic detection and classification of animal images is challenging, but it is much needed for many real-life applications. This paper presents a hybrid model of Mamdani Type-2 fuzzy rules and convolutional neural networks (CNNs) applied to identify and distinguish various animals using different datasets consisting of about 27,307 images. The proposed system utilizes fuzzy rules to detect the image and then apply the CNN model for the object's predicate category. The CNN model was trained and tested based on more than 21,846 pictures of animals. The experiments' results of the proposed method offered high speed and efficiency, which could be a prominent aspect in designing image-processing systems based on Type 2 fuzzy rules characterization for identifying fixed and moving images. The proposed fuzzy method obtained an accuracy rate for identifying and recognizing moving objects of 98% and a mean square error of 0.1183464 less than other studies. It also achieved a very high rate of correctly predicting malicious objects equal to recall = 0.98121 and a precision rate of 1. The test's accuracy was evaluated using the F1 Score, which obtained a high percentage of 0.99052.

**Keywords:** image classification; animal identification; convolution neural network CNN; Mamdani fuzzy rules; gaussian membership



## 1. Introduction

Object recognition is an essential and fundamental task in computer vision, which finds or identifies objects in digital images. Accurate and fast automatic detecting and classifying of animals' images is challenging, and it is much needed for many real-life applications [1]. The problem with most recognition models is that they require image objects without background and correct categorical labels. This enables the model to predict the right title of the item [2]. Image enhancement methods are used to solve the problem of impulsive noise in image processing. To control and eliminate animal diseases, farmers obtain assistance from identification and registration systems for monitoring all motion of the animals [3]. Feeding behavior is one of the activities in which their changes are commonly employed for predicting disease. When addressing the recognition process, the first important issue is arranging the class to be recognized. The organization of items at various levels has adopted techniques from the field of Cognitive Psychology. We can imagine a cat as a single object (cat), but it can also be identified as a four-legged, animated animal (quadruped). Cat is the term in a level semantic hierarchy, which easily comes to mind and does not in any way happen by chance or accident [4,5]. Experimental results present the existence of the main level in human classification. Over the decades, researchers put a lot of effort into other aspects of computer vision, with only a few developments in image recognition [6].

Therefore, recognizing an object in digital photos is not a trivial matter because of many challenges such as viewpoint shifts, scaling, background clutter, illumination changes, etc. It is considered a core task in computer-vision models [7]. PASCAL Visual

Object Classes (VOC) challenge and ongoing Image-Net wide Scale Visual-Recognition Challenge (ILSVRC) combine significant operations required for detecting answers that matter in video scenes. There is also increased interest in researching patient monitoring and healthcare [8,9].

The contribution of this paper is the design of a deep-learning framework for automatic object detection in a moving environment with a variable camera–object distance. For example, during experiments in a video environment, the objects move too far or too close to the camera, forcing the system to deal with the changing proportions of the image and its objects as a major factor in the detection process. The proposed design has four parts:

(1)　collecting the data,
(2)　accepting the command parameters,
(3)　defining the neural network model,
(4)　adjusting the model via training.

The algorithm ensures determining the details of the object or component in a single image during the movement and indicating the navigation within the video with reasonable time complexity. To the best of our knowledge, there has been no program or system with an integrated mechanism for identifying and discovering the components in an image within a video clip that takes into consideration the distance between the component and the camera as well as the movement of the camera during video shooting.

The practical application of the proposed system includes cases of studying animal behavior, especially in their wild environments, where it may be difficult to identify animals under study when animals make sudden movements that bring them very close to the camera or very far from it. The possible "customers" for the proposed animal image recognition system can be researchers and institutions, such as National Geographic, that work in Zoology and natural environments.

The scope of this work is to handle unpredictable movements in the process of animal image recognition, where these movements can bring the object either very close to the camera or very far from it. Such variations in camera–object distance can cause detection failure due to the variation of object proportions in the image.

This paper aims to build an adaptive model to detect and recognize animals based on hybrid deep learning, which helps obtain the shortest time and highest accuracy rate in identifying and classifying the objects. The model will deploy the Hybrid Mamdani fuzzy rules and convolutional neural network (CNN) to test two datasets consisting of several types of animals during detection and recognition phases.

This paper aims to build an adaptive model to detect and recognize animals based on hybridizing deep learning with an adaptive fuzzy system, which helps in obtaining the shortest training time and highest accuracy rate in identifying and classifying the objects. A combination of Mamdani Type-2 fuzzy system with adaptive rules with a Convolutional Neural Network CNN is applied to identify and distinguish various animals using different datasets consisting of about 27,307 images. The proposed system utilizes adaptive fuzzy rules to detect the image and then applies the CNN model for the object's predicate category. The model deploys the Hybrid Mamdani fuzzy rules and convolutional neural network (CNN) to test two datasets consisting of several types of animals during detection and recognition phases. The paper's structure is as follows: Section 2 presents the related work, while the third section focuses on unique details such as background and theoretical concepts. Section 4 reviews the dataset, and Section 5 will present details about the proposed work. The sixth section examines and esdiscuss the experimental work and results. Finally, Section 6 explains the conclusions of the proposed method.

## 2. Literature Review

Many researchers have discussed and reviewed the implementations of animals' image identification and classification using a variety of computing methods [10].

Gue [11] suggested automated face detection and individual identification for both videos and still-framed images using deep learning methods. The proposed system was

trained and tested with a dataset containing 102,399 images. The results show that the proposed method correctly identified the individuals with a rate of 94.1%, and it processed 31 facial images per second. Hou [12] presented the VGGNet model for identifying the giant panda and other animals' faces. The model was trained and tested on 65,000 face images of 25 pandas, which obtained a high accuracy rate of 95% for individual identification. Schofield [13] developed a face detection, tracking, and recognition model based on a deep convolutional neural network (CNN) for wild chimpanzees. The model was tested using datasets containing 10 million face images from 23 individuals. The proposed model achieved a high accuracy rate of 92.5% for identity recognition and 96.2% for sex recognition.

Nayagam [14] presented a method for recognizing objects of a video after detecting and extracting them. This was achieved by loading the video and extracting the frame, then catching the object of interest by implementing the global Label Distribution Protocol (LDP) features extracted, and sped-up robust features (SURF) detector. Ultimately, a comparison is made to the objects in the videos, which matched with the objects of intention. Leksut [15] addressed the problem of an object recognition model using a deep neural network technique to examine the effect of visual variation. They utilized iLab with 20M datasets of toy vehicle objects with variations of lighting, viewpoint, background, and focal setting. The experiment results on 1.751 million images from iLab. 20M showed significant improvement accuracy of object detection, (DenseNet: 85.6% to 91.6%; 86.5% to 90.71%), (AlexNet: 84.5% to 91.6%).

CNN improves variation learning and can notice special features and better learn object representations, decreasing the error rate of detection of ResNet by 33%, Alexnet by 43%, and DenseNet by 42. Kumar [16] suggested a hybrid model using a probabilistic neural network (PNN) and K-nearest neighbors (KNN) for classification of animal images. The proposed model consisted of several phases, such as segmentation, extraction, and classification. The experiments were deployed based on 4000 sample images for 25 classes of animals. The results showed that the KNN classifier achieved a high accuracy rate for 70% of samples compared to PNN in training (82.70, 78.09, respectively) using single block segmentation. Hendel [17] proposed an efficient method for detecting faces based on the Cambridge Face Memory Test (CFMT) and Cambridge Face Perception Test (CFPT). The method was tested and examined using fourteen individuals' performance factors. The results showed that they obtained better accuracy in face detection and face recognition than other studies. They achieved a 90% accuracy on the online version of CFMT. Chen [18] proposed the rudest tool for detecting and recognizing new object samples in the video by applying the Hybrid-Incremental Learning method (HIL) with Support Vector Machine. This can improve the recognition of unseen examples and concepts of the object by interacting with humans; the achieved hybrid technique improved the quality of recognition of k by reducing the error of prediction.

Another effort was made to remove the image's background for enhancing the work of the detection and recognition algorithm [19]. Boureau [20] suggested redesigning the object to remove the image's background and build a CNN model of object recognition that could accurately extract the image without its background.

The idea of fuzzy image segmentation is summarized as generating an adaptive fuzzy rule for segmentation to maintain a high accuracy level, where the process starts with a set of training images randomly selected from the available set of test images along with their original versions; then, a set of points is detected by Mamdani Fuzzy Rules within a rectangle around the region of interest in these training images. The remaining images are segmented using the initial fuzzy rules, and based on the quality of the result for any new image, the initial fuzzy rules are evolved; this process of rules updating is repeated every time there is a new image, to increase the segmentation accuracy. Results of adaptive segmentation will be the inputs for the Convolutional Neural Network, which adds a new level of adaptivity, hence a higher level of accuracy.

References [21,22] show details of Fuzzy Inference System Modeling and Fuzzy Inference System Tuning.

## 3. Materials and Methods

### 3.1. Methods

Machine learning is patterned studying that gives computers the capability of learning without using hardcoded values. Training of machine learning builds models and algorithms to predict new data. Figure 1 shows several algorithms in the machine learning model.

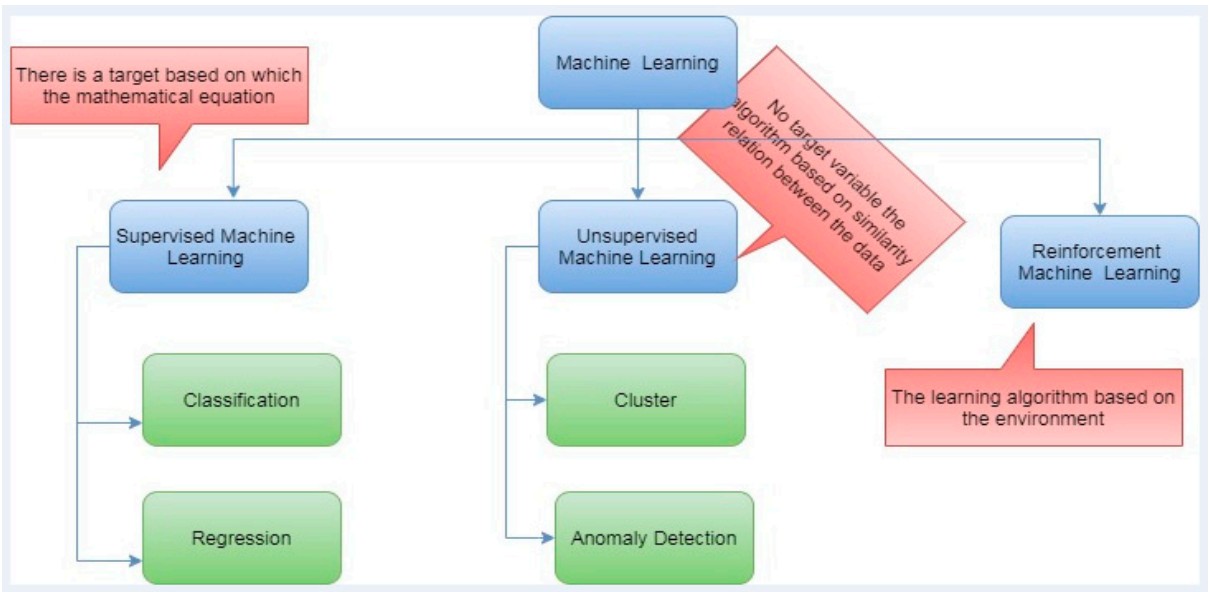

**Figure 1.** Machine learning techniques [23].

An important type of neural network in deep learning is CNN. It is specifically designed for image recognition and detection; it contains many layers of neural network that extracts features of an image and detects which class it belongs to (detection and recognition of an image after it is trained with a set of images). Figure 2 shows the architecture and working of CNN [24,25].

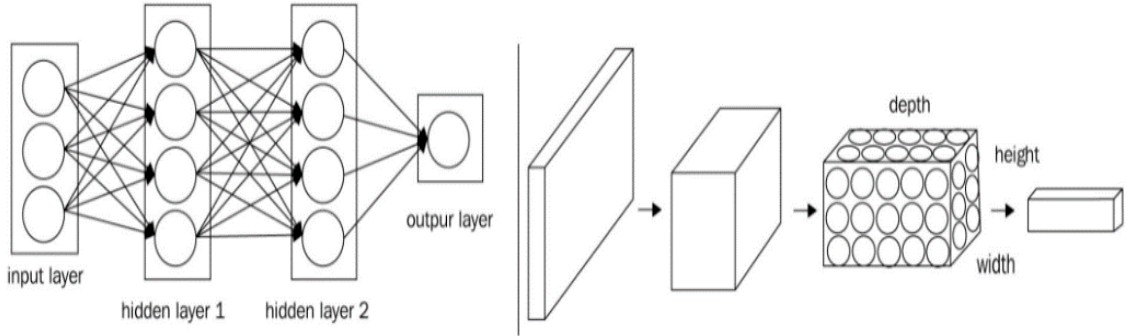

**Figure 2.** Architecture of the convolutional neural network (CNN).

A CNN contains layers of neural nets. In every layer, it converts one size of activations to others through a differentiable procedure. The three types of layers constructed by CNN are Convolutional Layer, Pooling Layer, and Fully Connected Layer. The first stage is the convolutional layer; it processes the image to extract only salient properties, filtering the input image and producing a feature map or activation map as illustrated in Figure 3 [23].

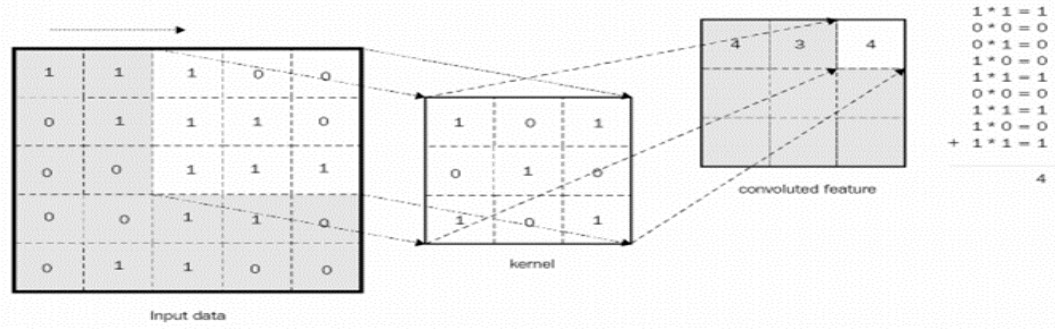

**Figure 3.** Convoluted Feature Vector with mask vector of size 3 * 3.

Convolution is a purely mathematical method achieved in three stages:

- The first stage is a sliding mask features and image matching patch,
- The second stage multiplies each input image pixel by the mask feature pixel,
- The third stage is summing all of them up and calculating the average of the results, and the final step is filling the results in a new matrix of features [26].

The CNN model pool process reduces the input image matrix's space using an appropriate mask size of 2 × 2 or 3 × 3 and applies the kernel function on every part of the image. Then the maximum value of results is selected [27]. The next stage in the structure of the CNN model is the rule function. This step obtains the pooling result where every pixel less than zero is made zero. The final step in the CNN model structure is the fully connected layer, which decides the input data label. This decision is based on the highest voted category [28]. The backpropagation learning algorithm reduces the error levels and produces the best prediction [29].

*3.2. Materials*

The experiment of the proposed work deploys two datasets of about 27,037 images downloaded from Kaggle for animals and resizes each image to 100 * 100 pixels as shown in Figure 4 [30,31].

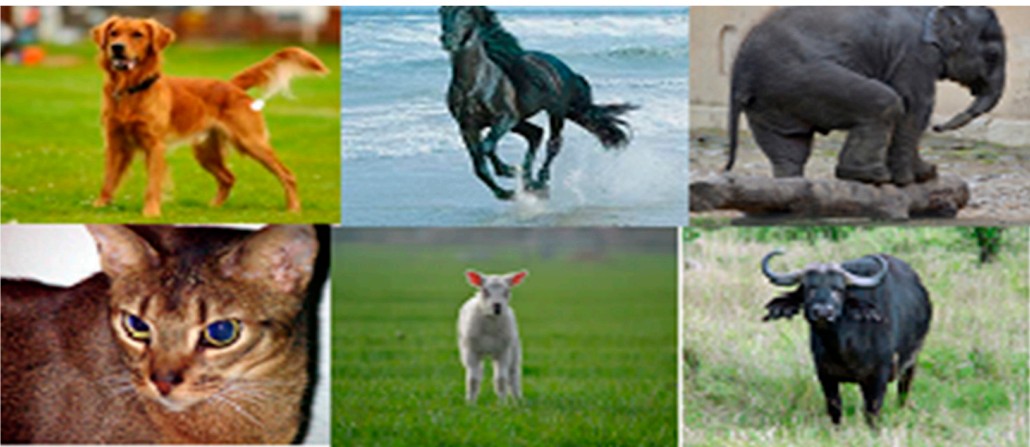

**Figure 4.** Sample of images from the datasets.

## 4. Proposed Work

The paper used many pictures collected from specialized researchers over the Internet, and an impressive collection of videos of animal pictures to show the path of movement, in addition to the well-known CIFAR-10 and CIFAR-100 databases which are labeled subsets of a dataset of 80 million tiny images collected by Alex Krizhevsky, Vinod Nair, and Geoffrey Hinton. The CIFAR-10 dataset consists of 60,000 32 × 32 color images in 10 classes, with 6000 images per class, with 50,000 training images and 10,000 test images. The

dataset is divided into five training batches and one test batch, each with 10,000 images. The test batch contains exactly 1000 randomly selected images from each class. The training batches contain the remaining images in random order, but some training batches may contain more images from one class than another.

### 4.1. Mamdani Fuzzy Rules for Edge Detection

Figure 5 shows the summarization of the proposed system. A detailed explanation is given below. Parts Fuzzy logic Type-2 depends on the Juzzy toolkit, whereas what remains of the code's parts is proprietary MATLAB code. Further details of Fuzzy Inference System Modeling and Fuzzy Inference System Tuning are given in [32].

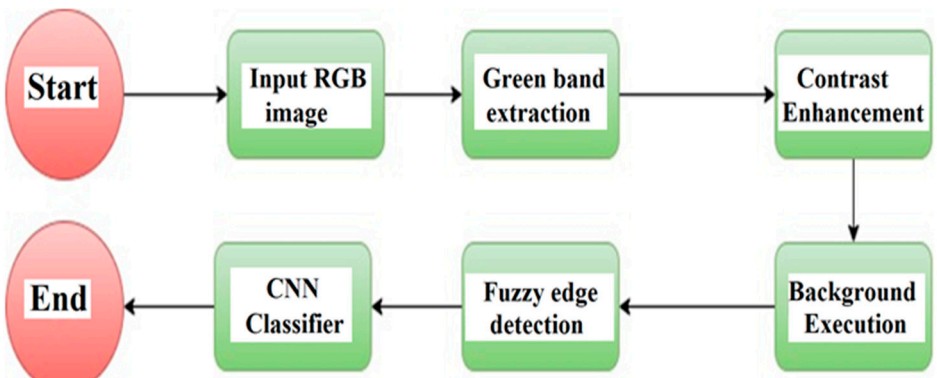

**Figure 5.** A flowchart of Mamdani fuzzy rules of the edge detection algorithm.

- Extracting green channel: The Red-Green-Blue images are transformed into green-channel pictures. Moreover, the green channel gives an intensity, which is higher [33] and extreme local contrast among the foreground and the background [34] compared to the other two channels.
- Contrast Enhancing: Contrast Limited Adaptive Histogram Equalization (CLAHE) is very efficient, with tiny non-intersecting areas named tiles in images. For each such tile, histogram equalization is applied. In the end, neighboring tiles are gathered using bilinear interpolation to eliminate boundaries induced artificially [35].
- Excluding Background: Background fluctuations in image luminance are removed so that foreground objects can be easy to analyze. Median filtering used with a kernel of 25 * 25 sizes is employed for blurring the image and smoothing the foreground. The background image information is removed by the process of subtracting image contrast-enhanced [Icontrasted] from the median filtered image [fmedian] as in Equation (1):

$$f' = \text{fmedian} - \text{Icontrasted} \tag{1}$$

where f' is the new image without the background.

The median filter substitutes the value of pixel f(x, y) by all pixels' median existing surrounding area neighborhood, as it is determined as in Equation (2) [36]:

$$\text{fmedian} (x, y) = \text{median} (i, j) \in \text{Wxy} \{\text{Iconstrasted}(i, j)\} \tag{2}$$

Where 'Wxy' indicates the area near and around the center (x, y) for image edge detection.

Fuzzy set {A} is a group of pairs that are arranged in a cumulative way formed of elements x of universal set X, degrees of corresponding membership µA (x) as presented in Equation (3):

$$A = \{(x, \mu A\ (x))\ |\ x \in X\}, \tag{3}$$

where µA (x) is a membership function taking values in a set that is arranged linearly of what named membership set in the interval [0, 1].

The proposed method utilizes the function of Gaussian membership in Equation (4):

$$\mu(x) = e\hat{\ }(-\{(x - m)\}^2/\{2k\}^2) \tag{4}$$

where m and κ represent the width and center of (fuzzy set {A}).

Fuzzy operations were carried out as two functions of Type_1 fuzzy rule, which are a footprint of Uncertainty, Lower Membership Function, and Upper Membership Function [37]. Among these two functions is the uncertainty area, where the proposed method selects the most appropriate parameter.

Membership function parameters are chosen by using multi-threshold Otsu [36] separately for images. As mentioned before in this topic, there are two linguistic variables; function input membership is a Gaussian function. The values of width (the function of Lower Membership), Lower and upper threshold values attained from the Otsu multi-threshold method are assigned to each language variable.

The function of Upper Membership values is chosen with an empirical method. Our implementation provides an extra set of tunable parameters, which depends on the Centroid-type for reduction algorithm of Type-2 Fuzzy Logic Systems (FLS) [38,39], as illustrated in Figure 6.

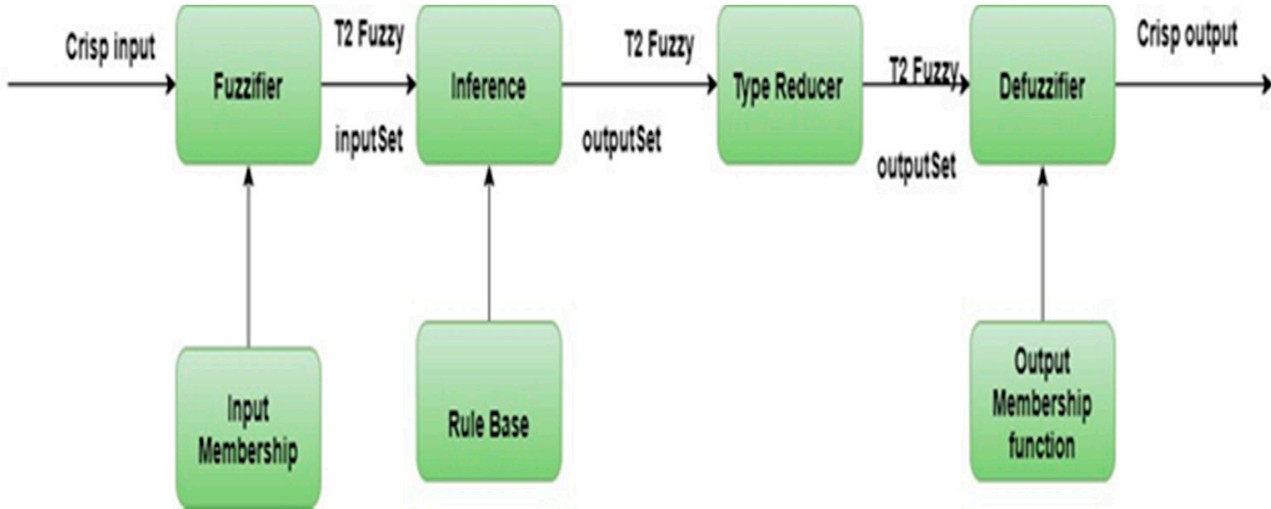

**Figure 6.** Type-2 system of the fuzzy logic system employed in the algorithm of STEAR.

As crisp inputs, four features are used, which can be listed as follows:

1.　Anti-diagonal Gradient Ik. In mathematics, an anti-diagonal matrix is a square matrix where all the entries are zero except those on the diagonal going from the lower-left corner to the upper-right corner (↗), known as the anti-diagonal.
2.　Diagonal Gradient Iz.
3.　Vertical Gradient Iy. Horizontal Gradient Ix.

The Fuzzifier utilizes Gaussian membership functions (Ix) with the linguistic variables as explained briefly in Table 1.

**Table 1.** Fuzzy membership functions input.

| Upper Membership Function | Membership Function | Lower Membership Function |
|---|---|---|
| Pixel: Black | Foreground | 0, 1 |
| Pixel: White | Background | 1, 0 |

The initial values of the Upper membership functions are chosen with an experimental method. In contrast, the lower membership function's initial values are determined separately for each image using a statistical approach. Moreover, the multi-threshold Otsu algorithm is employed for finding thresholds and utilized in the settings of the Lower membership function for functions of Input Membership.

In the same way, the other properties are configured based on the defined language variables. In this fuzzy system, the rules that define the edge are set to an Edge if all gradients' input values are white. Otherwise, it is specified as NO_Edge if all gradients' input values are black. The Upper membership function and Lower membership function's initial values were chosen in an experimental method, as illustrated in Table 2.

**Table 2.** Fuzzy input membership functions.

| Membership Function | Lower Membership Function | Upper Membership Function |
|:---:|:---:|:---:|
| Edge | 0.005, 0.035 | 0.04 |
| Not Edge | 0.99, 1.0 | 0.98, 1.0 |

Detecting Fuzzy edge is conducted automatically for each image depending on the Fuzzy Type-2 rules that choose the maximum gradient values. The two steps below illustrate the proposed fuzzy approach to find the edges of an image.

- Performs Gaussian blurring with a $3 \times 3$ kernel for an input image.
- Divide the blurred image into a $3 \times 3$ pixel matrix.

The eight directions of the convolution kernel are employed for calculating eight gradients in various directions, which decrease to four gradients. The Robinson operator possesses the eight convolution kernels to mirror the gradients, and the maximum mirror gradient is chosen as shown in Figure 7.

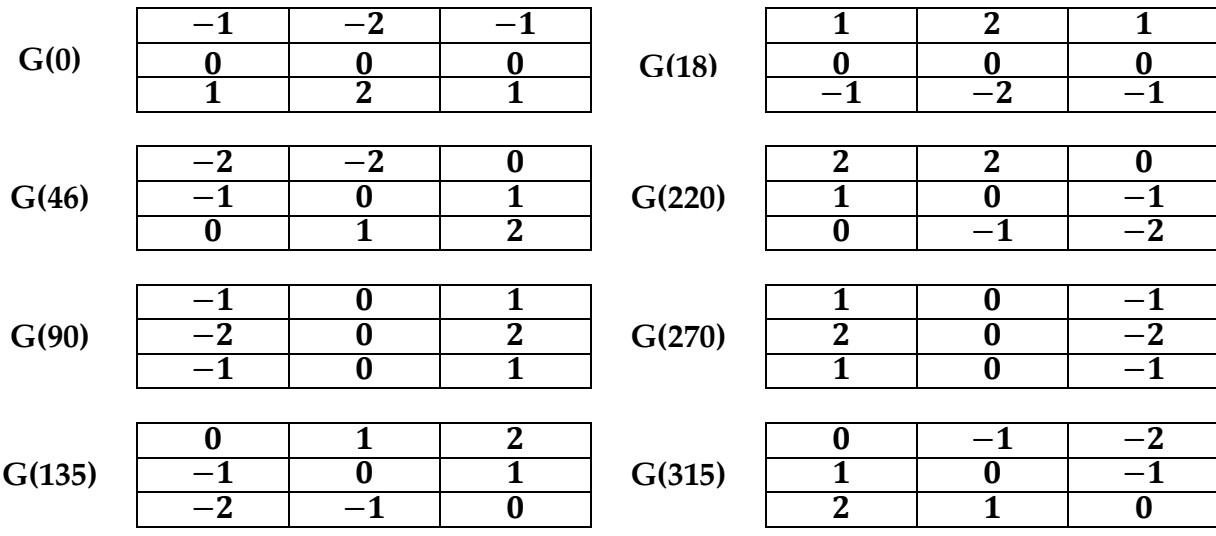

**Figure 7.** The mirrored kernels.

The existing two gradients for each row of the second convolutional core are compared, and the gradient with the maximum value is selected. Consequently, four matrices are attained rather than eight, which minimizes the fuzzy inference system implementation time. Ultimately, every gradient's value of a pixel is fed to a fuzzy inference system input for finding the edges. The locations with gaps were utilized using the post-filtration of the subsequent morphological closure and morphological discovery for combining vessels.

Kernels of sizes 8 × 8 and 2 × 2 are employed. The inverse binarization of an image I(x,y) is illustrated in Equation (5):

$$I_b(x,y) = 255 \, [1 + sign\{I(x,y) - threshold\}]/2 \tag{5}$$

where sign(.) is the MATLAB signum function. Hence, a binary image is created by setting a threshold on the pixel intensity of the original image. The thresholding process is used for separating the image into foreground values (black) and background values (white).

### 4.2. Recognition of Moving Objects Using CNN

After edge detection and feature extraction by applying the Mamdani fuzzy rules, CNN is used to train and build a model capable of predicting the new object. CNN can simulate and build a model that classifies the color images; for example, when a green color image is 100 * 100 pixels, then the CNN model's hidden layer is designed to be 100 * 100 * 1 = 10,000 Weight. Then, the CNN input solves the scalability problem. The layers of CNN possess neurons arranged in three dimensions (height, depth, and width), as demonstrated in Figure 8.

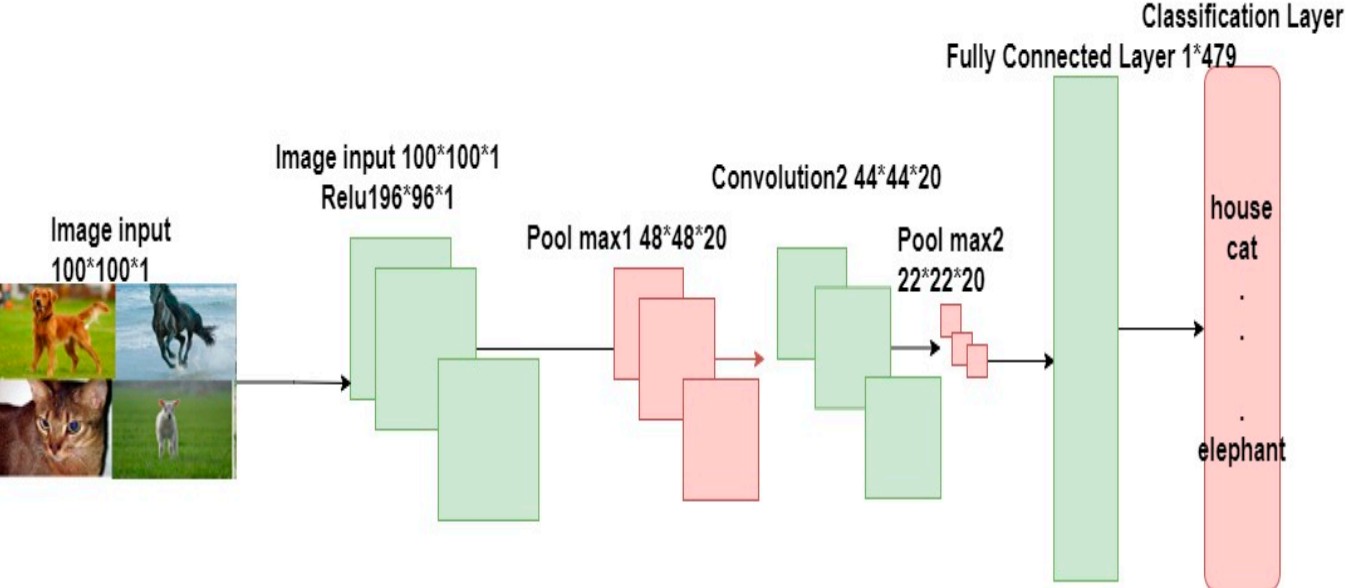

**Figure 8.** Diagram of the proposed CNN.

## 5. Experiments and Results

Two datasets consisting of 27,037 images were used and tested in the current work. They were downloaded from the Kaggle website, and each included an animal image. The proposed work is composed of two basic steps. The first step is the detection and feature extraction of the object in the removed background by applying the Mamdani fuzzy rules. The second step is implementing the CNN to recognize the item.

### 5.1. Edge Detection and Feature Extraction

An example of results attained by using the method proposed and other methods employed for making the comparison is shown in Figure 9. Traditional approaches to determine boundaries such as Robert, Prewitt, FreiChen, Laplacian, Sobel, Canny, and Robinson were substandard to the Fuzzy Edge detection that is proposed. Vessels of blood are "broken" in images processed by algorithms that rely on FreiChen, Sobel, and Prewitt operators. Algorithms that depend on Canny and Robinson operators were significantly better.

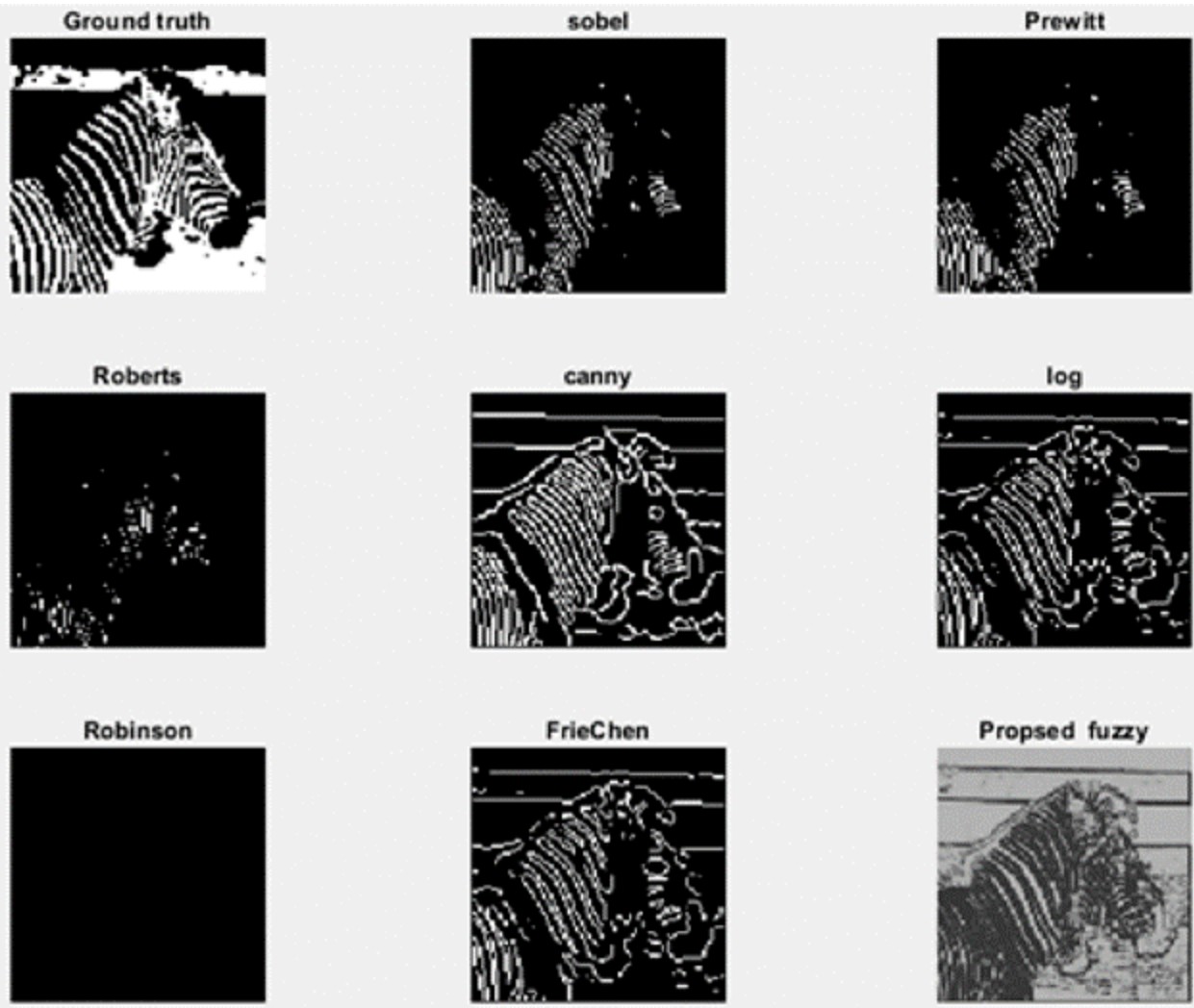

**Figure 9.** The result of comparing image edges.

This result is related to other traditional methods to determine boundaries, but images have shown that specific animals were disconnected. The fuzzy method delivered better results than the Prewitt factor, Robert, and Laplacian operators. Canny and Robinson's operators had a good result visually and showed fewer results than fuzzy logic. The comparison of obtained results was evaluated using the following performance measures: peak signal-to-noise ratio (PSNR), signal-to-noise ratio (SNR), mean-squared error (MSE), and structural similarity index (SSIM) [40–42], as illustrated in Table 3 and Figure 10.

**Table 3.** Evaluation metrics summary.

|  | PSNR | SNR | MSE | SSIM |
|---|---|---|---|---|
| Ground truth | 6.44359723822 | 0.390523 | 0.1474757 | 0.04262958446 |
| Sobel | 6.40926093734 | 0.047160 | 0.1486463 | 0.0372840370 |
| Prewitt | 6.40886536226 | 0.043204 | 0.1486598 | 0.037262112483 |
| Roberts | 6.40575761207 | 0.012127 | 0.1487663 | 0.037137089 |
| Canny | 6.41505149913 | 0.105066 | 0.1484482 | 0.0393885 |
| Log | 6.4045448 | 9.64327 | 0.1488078 | 0.037097395897 |
| Robinson | 6.41454242 | 0.099975 | 0.1484656 | 0.038573626127 |
| FrieChen | 6.41454242 | 0.099975 | 0.1484656 | 0.03857362612 |
| Proposed fuzzy | 7.7161813 | 0.307274 | 0.1183464 | 0.784321825 |

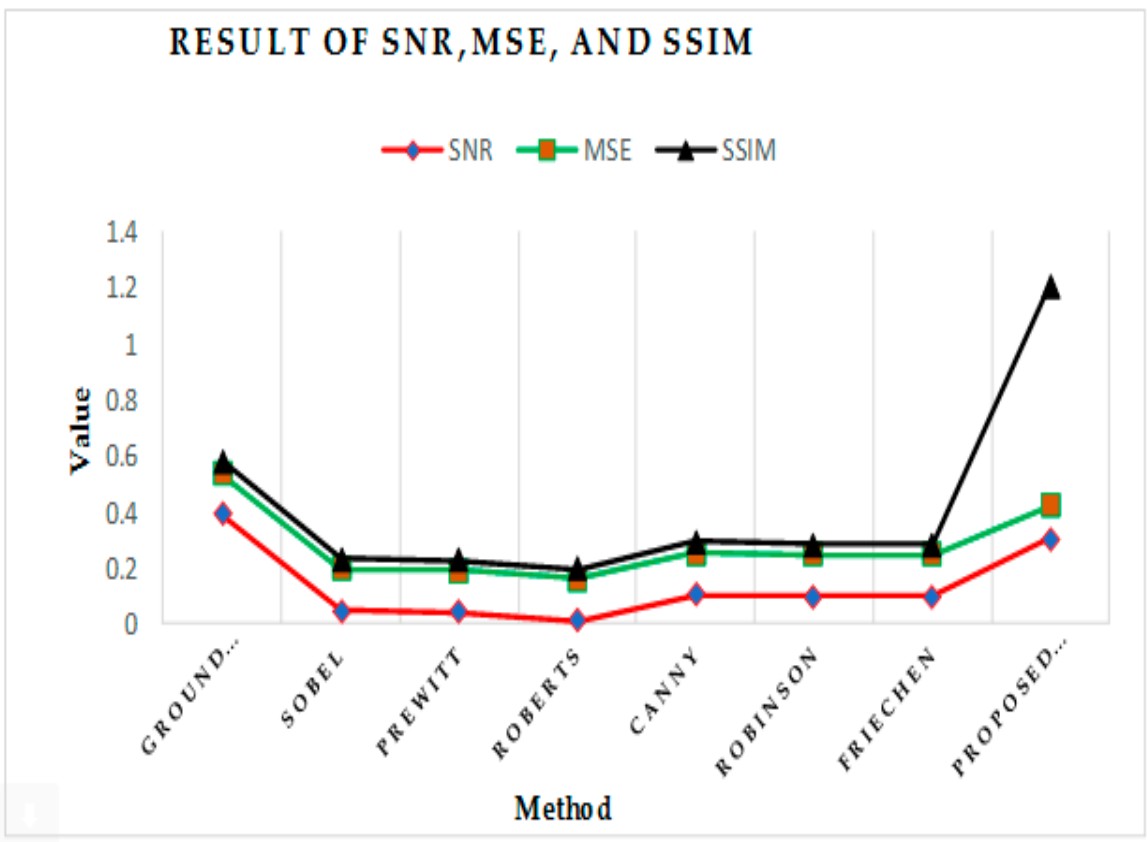

**Figure 10.** Performance evaluation factors using signal-to-noise ratio (SNR), mean-squared error (MSE), structural similarity index (SSIM).

### 5.2. The Recognition of a Moving Object by CNN

After detecting an object's location and removing the background of the frame, the CNN training model was introduced. Then, the dataset was split into 80% training set (21,846 images) and 20% testing set (5461 images) used by the cross-validation process. The entered object's size was 100 * 100 * 3, and the training setup of CNN worked, as shown here. The rate of learning was 0.001, and momentum was 0.9. The network's architecture was composed of four convolutional layers and four pooling layers, which were followed by two connected layers.

The rectified linear unit (ReLU) activation function was implemented in the hidden layers as a sparse regularization convolutional neural network to improve CNN's performance. The ratio of dropout was set to 0.5, the rate of learning initially set to 0.001, and the training was paused after 1000 epochs, as illustrated in Table 4.

The results of training the hybrid Mamdani fuzzy and CNN network are shown in Figure 11. In epoch 1, the Elapsed time parameter was set to 2 s, and the achieved rate of accuracy was 28.13%, while the parameter of mini-batch loss was 1.4149. Then, at epoch 41, the accuracy rate reached 100%. The parameter of mini-batch loss was 0.0021, and the elapsed time was 16.32 min.

**Table 4.** Analysis result of the CNN model.

| Layer | Name | Activations | Learnable | Properties |
|---|---|---|---|---|
| 1 | Image input | 100 * 100 * 1 | - | The zero-center normalization approach |
| 2 | Convolution1 | 96 * 96 * 1 | Weight 5 * 5 * 3 * 20Bias 1 * 1 * 20 | The convolution mask size is 5 * 5 with 20 filter and padding [ 0 0 0 0] and stride [1 1] |
| 3 | Relu1 | 96 * 96 * 1 | - | Relu1(x) = max(0,x) |
| 4 | Pool max1 | 48 * 48 * 20 | - | The pooling is tacking the max value in window2 $\times$ 2 with padding [0 0 0 0] and stride [2 2] |
| 5 | Convolution2 | 44 * 44 * 20 | Weight 5 * 5 * 20 * 20Bias 1 * 1 * 20 | The convolution mask size is 5 * 5 with 20 filter and padding [ 0 0 0 0] and stride [1 1] |
| 6 | Relu2 | 44 * 44 * 20 | - | Relu2(x) = max(0,x) |
| 7 | Pool max2 | 22 * 22 * 20 | - | The pooling is tacking the max value in window2 $\times$ 2 with padding [0 0 0 0] and stride [2 2] |
| 14 | Fully Connected Layer | 1 * 1 * 5 | - | Multiplies weight matrix by an input image and then adds a bias vector. |
| 15 | SoftMax Layer | 1 * 1 * 5 | - | activation function |
| 16 | Classification Layer | - | - | Calculates the cross-entropy |

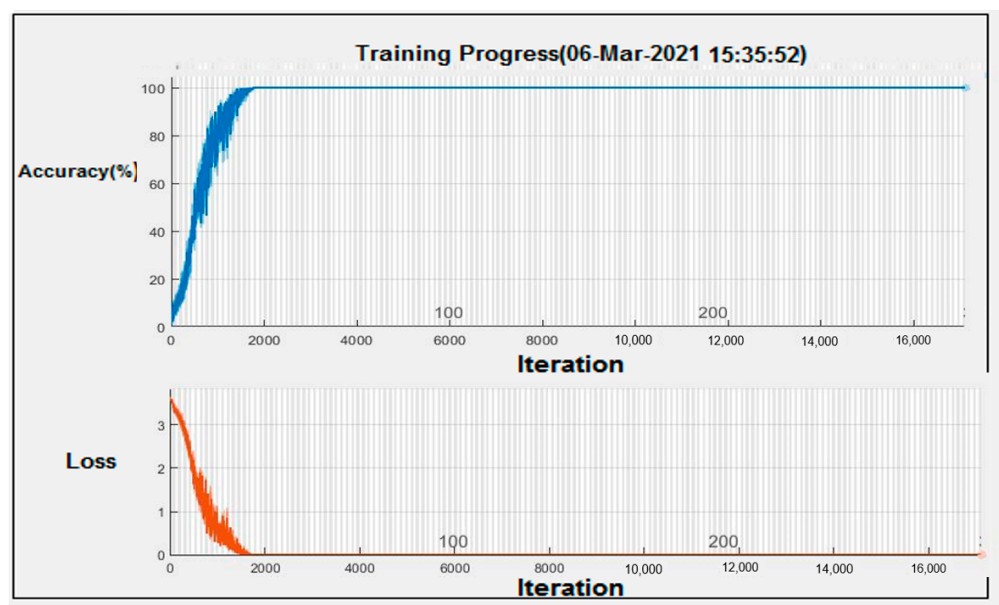

**Figure 11.** Training the proposed hybrid Mamdani fuzzy and CNN network.

The results of training the CNN network are shown in Figure 12. In epoch 1, the Elapsed time parameter was set to 2 s, and the achieved rate of accuracy was 25.78%, and

the parameter of mini-batch loss was 1.3868. Then, at epoch 73, the accuracy rate reached 100%. The parameter of mini-batch loss was 0.0172, while the elapsed time was 27.35 min.

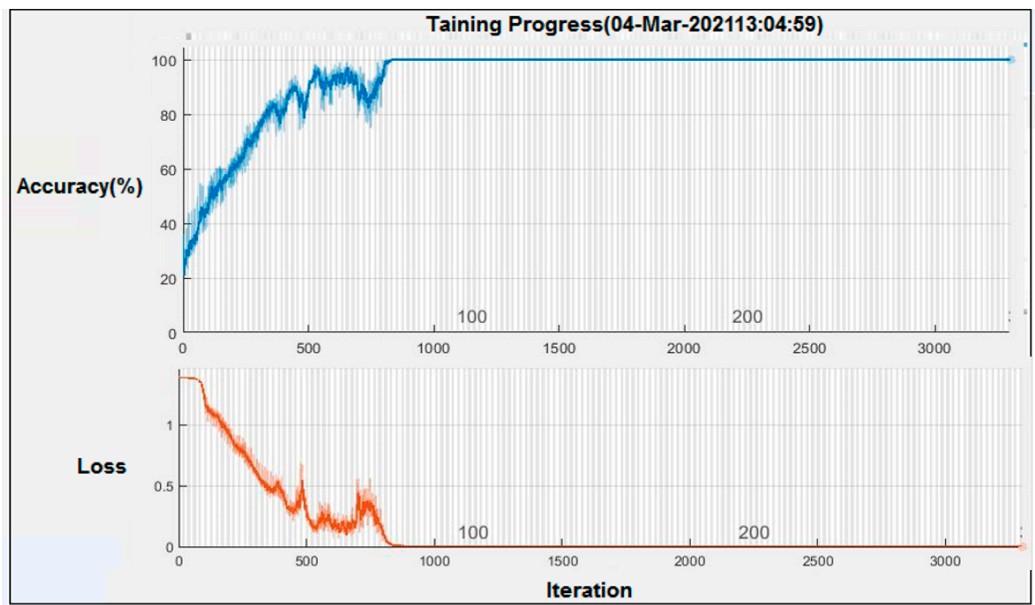

**Figure 12.** Training the CNN network.

Comparing the hybrid fuzzy-CNN and CNN models shows that the proposed approach is more efficient than the traditional CNN model. The proposed system was able to reach 100% in less time than the traditional CNN by 11.0300. In addition, the conventional experiment's mini-batch loss rate was higher than the proposed hybrid Fuzzy–CNN model by 0.0151. The hybrid Fuzzy–CNN model reached 100% accuracy at an epoch of 41, which is less with 32 epochs in the traditional CNN. Figure 13 shows the result of recognition samples using the hybrid Fuzzy–CNN with Mamdani fuzzy rule.

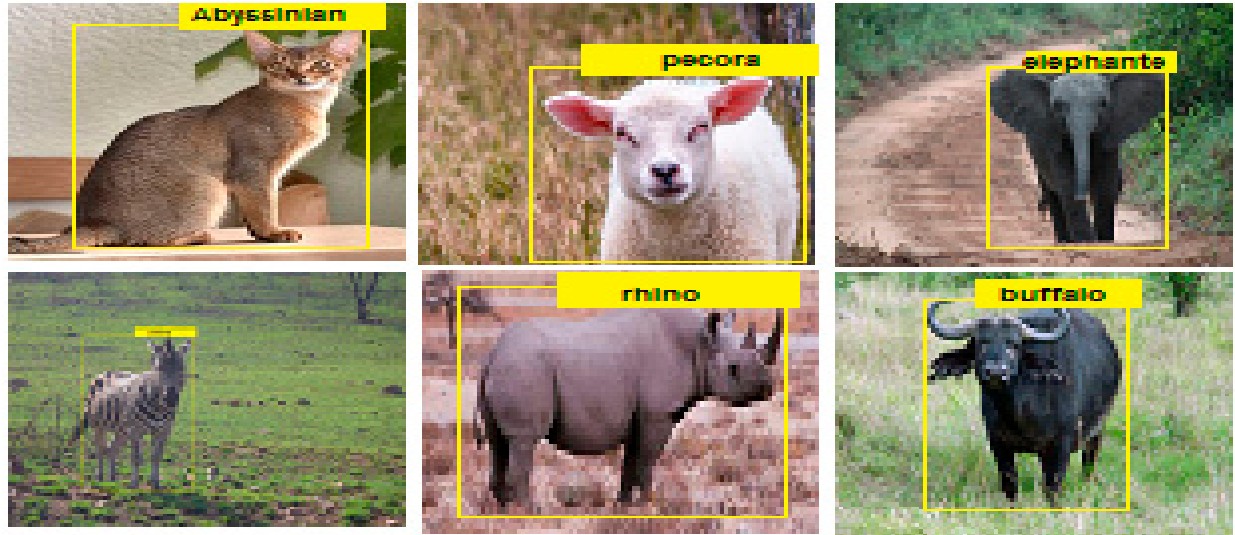

**Figure 13.** Result of recognition samples using the hybrid CNN with Mamdani fuzzy rule.

Due to the difficulty of having a fully integrated system for identifying and manipulating objects accurately, this paper has focused on the object's proximity to the camera in the case of rapid movement of animals while taking pictures of them or shooting a video of the movement path; the aim is that the detection and recognition of the moving object in

the video is performed accurately irrespective of the distance between the camera and the image component.

This work has carefully selected a dataset that contains moving animals and humans in the CIFAR-10 database Google net, where each image is 100 * 100 pixels.

Figure 14 shows how accurate the proposed approach is in identifying objects in video images under extreme camera–object distances.

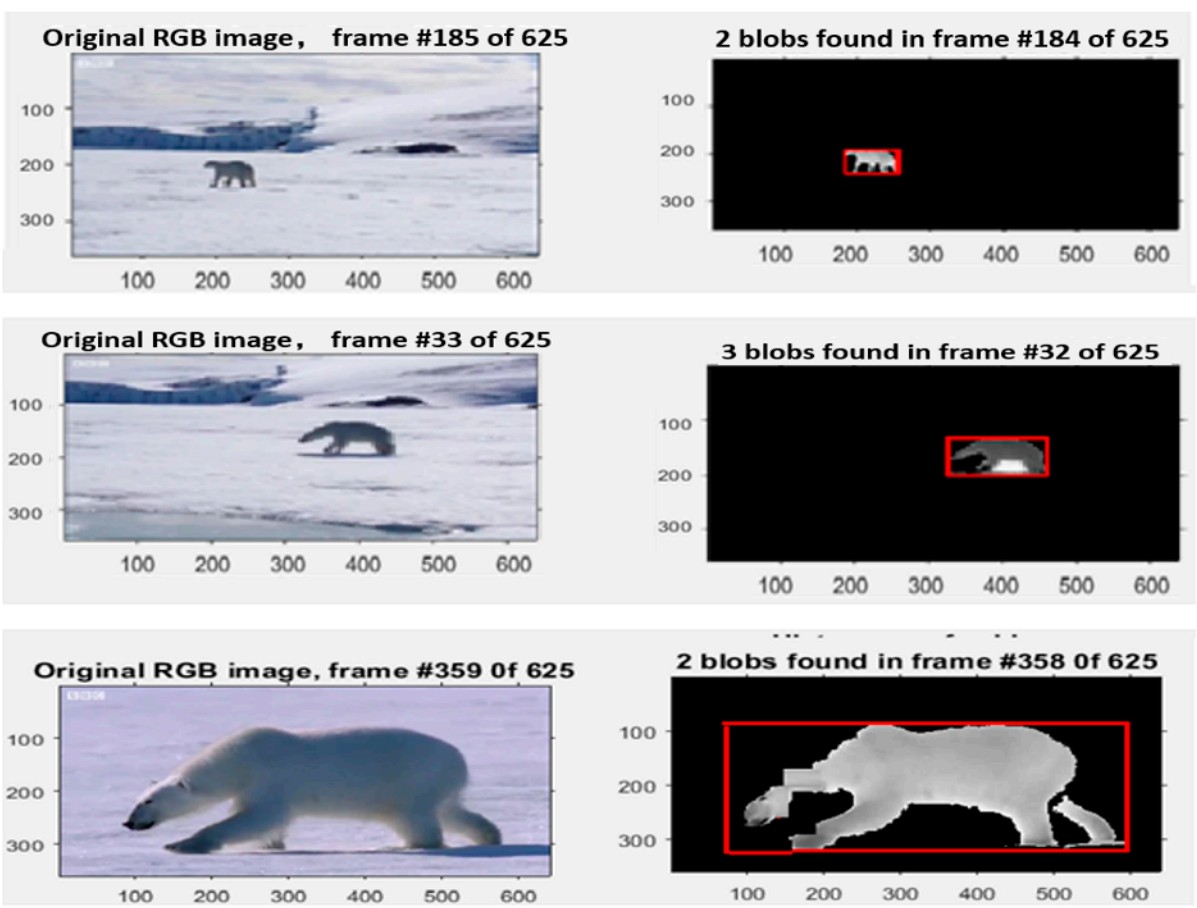

**Figure 14.** Accuracy of identifying objects in video images under extreme camera–object distances using the proposed approach.

### 5.3. Comparison of Results

In Table 5, a detailed classification of the test samples is listed. The "True" and reference columns represent the "True" situation, while the row values are the predicted "True". Table 6 contains the statistical measures of the test phase. Comparing the four models show that the CNN has high accuracy. The proposed system results have been compared to the results of six other related works, and it shows that the proposed system is more accurate than these six studies, as shown in Table 7.

**Table 5.** Results of Classification process.

| Model Type | Prediction | Reference | |
|---|---|---|---|
| | | Positive | Negative |
| CNN | True | 26,794 | 0 |
| | False | 0 | 513 |

**Table 6.** Test Phase Statistic Measures results.

| Statistic | Description | CNN + Kalman Filter |
|---|---|---|
| Accuracy | Rate of correctly predicted<br>ACC = TP + TN/(TP + TN + FP + FN) | 0.98121 |
| True positive | Number of correctly predicted. | 26,794 |
| True Negative | Number of wrong objects which are correctly classified | 0 |
| False positive | Number of incorrectly predicted | 0 |
| False Negative | Number of wrong objects which are incorrectly predicted | 513 |
| Misclassification Rate | The percentage of incorrectly predicted = (FP + FN)/total | 0.018786 |
| Specificity | calculated as the number of correct negative predictions Specificity = TN/(TN + FP) | NaN |
| Precision | Calculated as the number of correct positive Precision = TP (TP + FP) | 1 |
| Sensitive Recall | Rate of correctly predicted malicious objects SensitiveRecall = TP/(TP + FN) | 0.98121 |
| F1_Score | Measure of accuracy of test. It considers the precision p and the recall (R) of the test for computing the score F1_Score = (2 * (Sensitive_Recall * Precision))/(Sensitive_Recall + Precision) | 0.99052 |

**Table 7.** Comparison with some related works.

| Authors | Tools | Accuracy |
|---|---|---|
| This proposed work | Mamdani fuzzy rules + CNN | 98.121% |
| Muhammad [38] | SVM classifier | 97.95% |
| Bruno [39] | ANN and k-NN | 97% |
| Tian [40] | Hierarchical Filtered Motion | 94%KTH Human Action Dataset |
| Kumar [41] | Gabor-Ridgelet Transform | 96% KTH Human Action Dataset |

## 6. Conclusions

This paper proposed an accurate, fast, and automatic method for detecting and classifying animals' images for both fixed and moving pictures. A hybrid model of Mamdani Type-2 fuzzy rules and Convolutional Neural Networks CNN was applied to identify and classify various animals using about 27,307 images, where the proposed system was trained based on more than 21,846 pictures of animals and was tested using 5461 images (80% training, 20% testing). The challenge was to build an automatic CNN model for detection and recognition of animals on a different dataset with a huge number of images. This challenge has been addressed in this work by a novel, double-adaptive hybrid model using Mamdani Type-2 fuzzy rules and CNN. In this model, the fuzzy system removes the unnecessary information to reduce the number of CNN layers, which results in less training time. More accurate classification is obtained due to the double adaptivity. The experimental results of the proposed method offered high performance with less time-complexity than other studies in related works. The proposed fuzzy method obtained an accuracy rate for identifying and recognizing moving objects of 98% and a mean square error less than other studies. It also achieved a very high rate of correctly predicting wrong objects.

**Author Contributions:** Conceptualization, H.R.M.; methodology, H.R.M.; software, H.R.M., Z.M.H.; validation, H.R.M., Z.M.H.; formal analysis, H.R.M.; investigation, H.R.M.; resources, H.R.M., Z.M.H.; data curation, H.R.M., Z.M.H.; writing—original draft preparation, H.R.M.; writing—review and editing, Z.M.H.; visualization, H.R.M.; supervision, Z.M.H.; project administration, Z.M.H.; funding acquisition, H.R.M., Z.M.H. All authors have read and agreed to the published version of the manuscript.

**Funding:** This research received no external funding.

**Data Availability Statement:** Data is contained within the article.

**Acknowledgments:** The authors acknowledge University of Kufa, Computer Science and Mathematics, for providing high-performance computing resources and expertise to support this research.

**Conflicts of Interest:** The authors declare no conflict of interest.

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
