# Peer review of "Hybrid Mamdani Fuzzy Rules and Convolutional Neural Networks for Analysis and Identification of Animal Images"

_computation, doi:10.3390/computation9030035_

Round 1
Reviewer 1 Report
In this paper, the authors proposed a CNN-based method for animal image classification. I have the following comments and suggestions
- The novelty of this paper is quite limited. The authors should highlight the contributions of this paper.
- The proposed method is quite similar to the current existing methods and it is significant for authors to release the codes and dataset of this paper so that readers can easily follow this work.
- Please the authors clarify how to collect and process data and whether is a significant contribution of this paper.
- Please the authors indicate how to access the dataset.
Author Response
Dear reviewer
We have revised our article according to your comments. The attached file has a detailed response to the received comments. We want to take this opportunity to thank the editorial office and the reviewers for their valuable comments and informative feedback, which enriched the article.

Reviewer 2 Report
I was very pleased for me to review you paper.
The hybrid method you propose is interesting and original. I hope you will spread this knowledge to other areas connected with image recognition.
Unfortunately, you did not pay attention in the introduction to the scope of the proposed solution. It is understandable that your solution applies to animal image recognition. However, what is the practical application? Where are the specific "customers" for your future animal image recognition system? There is no answer to this question in the article.
I also draw your attention to Figure 8 - it is a slightly modified figure that has been published many times (https://machinelearning.mipa.ugm.ac.id/2018/06/25/fully-connected-layer-cnn-dan -implementasinya /) (https://www.apriorit.com/dev-blog/603-action-detection-using-dnn). However, you do not provide links to sources (in the case of Figures 2 and 6, I have no questions, since there are links). I don't know if the original image of Figure 8 is under copyright, but it will be better to make your own image.
Author Response

(The authors gave the same response as above.)

Reviewer 3 Report
Authors suggest the use of fuzzy rules to improve CNN classification performance.
The approach is interesting, however, the article needs strong improvements.
-Introduction should explain the problem, and should justify the use of machine learning techniques for this specific problem. Why do you need to classify animals? Moreover, a brief description of the proposed method should be introduced here. It it not clear, indeed, if the authors are proposing a new method, or a pipeline of existing methods. Please highlight which are the main findings of this paper
- Literature review should be a critical description of the state-of-the-art. It should explain the limitations and how the proposed approach is able to overcome them. It is not just a sequential description of methods. Moreover, literature in the article focuses on CNN references, while works on fuzzy image segmentation are ignored. Please discuss how fuzzy systems have been used in literature for image segmentation (e.g. https://doi.org/10.1016/j.ins.2014.06.025 ) and how these methods differ from your proposal.
- Methods: author must detail here the pipeline that are suggesting, otherwise the main contribution of this paper is not clear. Mamdami fuzzy Rules must be described, since the article should be clear also to not expert users. Instead of an image with the taxonomy of machine learning methods (figure 1), a schema of the proposed approach is needed here. (Please, when use third part images add a reference. e.g. Figure 2 has been cut from somewhere, indeed there is still part of the page numbering in the top left corner.)
- paragraph 3 is called materials and methods, but it contains only the description of the methods. Data description must be added here, or since you are using benchmark datasets, they could be described in the experimental part, but the title of the paragraph should be changed
- Paragraph 4. The method is not clear. All the variables used in the equations must be explained. Please detail all the steps of the pipeline in figure 5. Moreover, it would be useful visualizing fuzzy sets and fuzzy rules that have been derived. Please add references/links to the libraries/algorithms that have been used. It is not clear which is the novelty of the proposed approach. Is the use of this pipeline for this specific dataset? Does it work with this data only? Is it a generic framework for image segmentation and classification? It is not clear which are the parameters authors refer to at page six, row 208. Finally, figure 6 must be definitely better explain.
- Experimental set must be described. Did you use cross validation? How did you split data in training and test set? Please describe the evaluation measures that have been used, otherwise it is not possible to interpret the results. Paragraph should firstly introduce all what will be done, and then results should be described. For example, paragraph 5.1 shows a comparison among different edge detection methods, but this was not expected. Please add references to the methods that have been reported here. Moreover, figure 9 shows that, while all the methods use white color for the edges and black for the background, the fuzzy approach inverts the color. Does this impact the results? Again, the evaluation metrics must be described. Extended names must be used before acronyms (PSNR, SNR, etc.)
- I don't know the meaning of PSNR and SWIVEL metrics, but it is clear that MSE value of your method is higher than those of the other methods, thus you collect a higher error. Please discuss this issue.
- Table 5 and 6: please use tables and not log images.
- Table 7 shows a 100% accuracy. This is commonly due to overfitting. Did you used different sets of images for training and testing your algorithm?
- Table 8: please correct the measures definitions.You refer to malicious objects that are not related to this problem. Moreover other definitions are not correct. True positive are not the number of correctly predicted....what?
- Table 9 shows a comparison with other works in which other methods have been used. It is not clear if the same data has been used and the same experimental settings. Otherwise results are not comparable.
Author Response

(The authors gave the same response as above.)

Round 2
Reviewer 3 Report
Authors have improved the article according with the reviewers' suggestions. Please use high quality images, and resize them in order to correctly show the text inside them (e.g. fig 4 6 and 8).
This manuscript is a resubmission of an earlier submission. The following is a list of the peer review reports and author responses from that submission.